# New Insights into Arrestin Recruitment to GPCRs

**DOI:** 10.3390/ijms21144949

**Published:** 2020-07-13

**Authors:** Martin Spillmann, Larissa Thurner, Nina Romantini, Mirjam Zimmermann, Benoit Meger, Martin Behe, Maria Waldhoer, Gebhard F. X. Schertler, Philipp Berger

**Affiliations:** 1Paul Scherrer Institute, Laboratory of Nanoscale Biology, PSI, Forschungsstrasse 111, 5232 Villigen, Switzerland; Martin.Spillmann@psi.ch (M.S.); larissa.thurner@online.de (L.T.); Benoit.Meger@psi.ch (B.M.); 2Paul Scherrer Institute, Center for Radiopharmaceutical Sciences (CRS), PSI, Forschungsstrasse 111, 5232 Villigen, Switzerland; nina.romantini@outlook.com (N.R.); Martin.Behe@psi.ch (M.B.); 3InterAx Biotech AG, PARK innovAARE, 5234 Villigen, Switzerland; zimmermann@interaxbiotech.com (M.Z.); waldhoer@interaxbiotech.com (M.W.); 4Paul Scherrer Institute, Division of Biology and Chemistry (BIO), PSI, Forschungsstrasse 111, 5232 Villigen, Switzerland; Gebhard.Schertler@psi.ch

**Keywords:** GPCR, arrestin, receptor activation, signaling

## Abstract

G protein-coupled receptors (GPCRs) are cellular master regulators that translate extracellular stimuli such as light, small molecules or peptides into a cellular response. Upon ligand binding, they bind intracellular proteins such as G proteins or arrestins, modulating intracellular signaling cascades. Here, we use a protein-fragment complementation approach based on nanoluciferase (split luciferase assay) to assess interaction of all four known human arrestins with four different GPCRs (two class A and two class B receptors) in live cells. Besides directly tagging the 11S split-luciferase subunit to the receptor, we also could demonstrate that membrane localization of the 11S subunit with a CAAX-tag allowed us to probe arrestin recruitment by endogenously expressed GPCRs. Varying the expression levels of our reporter constructs changed the dynamic behavior of our assay, which we addressed with an advanced baculovirus-based multigene expression system. Our detection assay allowed us to probe the relevance of each of the two arrestin binding sites in the different GPCRs for arrestin binding. We observed remarkable differences between the roles of each arresting binding site in the tested GPCRs and propose that the distinct advantages of our system for probing receptor interaction with effector proteins will help elucidate the molecular basis of GPCR signaling efficacy and specificity in different cell types.

## 1. Introduction

Arrestins are cytosolic adaptor proteins, capable of binding to transmembrane proteins, most notably, G protein-coupled receptors (GPCRs). They obtained their name from their ability to desensitize GPCRs by arresting the heterotrimeric G protein signaling pathway. Vertebrates have four arrestin genes: S-arrestin (*SAG*), arrestin-C (*ARR3*), β-arrestin-1 (*ARRB1*) and β-arrestin-2 (*ARRB2*) [1] S-arrestin and arrestin-C are also referred to as visual arrestins, due to their selective expression in rod and cone photoreceptor cells of the retina where they are involved in the desensitization of rhodopsin and cone opsins [2]. β-arrestin-1 and β-arrestin-2, also called non-visual arrestins, are expressed throughout the body and bind to a great variety of GPCRs. These receptors can be classified into two groups in terms of their interaction with β-arrestins: class A, which interact only transiently with arrestins and show a preference for β-arrestin-2 (like the β2-adrenergic receptor (B2AR), and class B, which have a prolonged interaction and no apparent selectivity (like the type-2 angiotensin II receptor) [3].

Besides desensitization of G protein signaling, arrestins also act as multifunctional scaffold proteins. They are thus involved in endosomal trafficking, vesicle sorting and signal modulation [4]. Internalization and trafficking of GPCRs is initiated by binding of clathrin heavy chain and adaptor protein AP2 [5]. It has been shown that internalized receptors still contribute to signaling [6] Arrestins are also scaffolding modules for elements of the MAP kinase pathway, including c-Jun N-terminal kinase (JNK), ERK1/2 and p38 MAPK. [7]. Further, arrestins have been shown to interact with c-Src, Ca^2+^-bound calmodulin, microtubules and E3 ubiquitin ligases. The interaction with ubiquitin ligases plays an important role in the targeting of internalized receptors for degradation, leading to desensitization [8].

The GPCR–arrestin interaction has long been used as an indirect measure of ligand binding, not only in basic research but also in the pharmaceutical industry. Therefore, many assay strategies have been developed, and various arrestin-receptor interaction assays have been employed, for example, PathHunter, Tango or assays based on bioluminescence resonance energy transfer (BRET).

The Tango assay is based on a fusion protein of TEV protease with arrestin, releasing a transcription factor (TF) from a GPCR-TF fusion protein if they are in close proximity after activation. This leads to expression of a reporter gene in the nucleus. It is an endpoint measurement taken after overnight incubation with a ligand, meaning that it does not reflect the typical arrestin recruitment, which occurs within minutes [9]. The PathHunter assay also requires the expression of a modified β-arrestin-2 and a GPCR, which is C-terminally tagged with a β-galactosidase enzyme fragment [10]. Both Tango and PathHunter assays have been proven to be useful for high-throughput screening of GPCR ligands in the pharmaceutical industry [11,12]. Membrane recruitment of a GFP-tagged arrestin after GPCR stimulation with a ligand can also be quantified using classical fluorescence microscopy and an image analysis software (Transfluor assay). Nevertheless, it shows merely internalization of arrestin into endosomes, not direct interactions [13].

BRET assays allow time-resolved measurements of signaling events. BRET assays utilize a quantum mechanical effect that allows the transfer of energy from a light-emitting donor to an acceptor fluorophore, given a certain spectral overlap. As the efficiency of this transfer is highly distance-dependent, it can be used to study protein–protein interactions in living cells in real time. It was initially used to measure GPCR dimerization, and later to measure recruitment of a β-arrestin2–YFP fusion protein to B2AR fused to *Renilla* luciferase (B2AR–Rluc) [14,15]. Recently, BRET assays were used to study the pluridimensional efficacy (or bias) of ligands, where each pathway activated by the GPCR is measured independently to gain a more holistic understanding of the signaling profile created by a ligand [16].

Alternatively, intramolecular arrestin BRET sensors can be used for monitoring conformational changes, thereby indirectly measuring activation and recruitment to a receptor. For example, β-arrestin-2 was fused to *Renilla* luciferase and yellow fluorescent protein at its N- and C-termini, respectively [17]. Another approach uses FlAsH-BRET, where a tetracysteine motif that binds to a fluorescent dye was incorporated into β-arrestin-2. The motif was incorporated at different positions to report different structural changes. A luciferase fused to the β-arrestin-2 N-terminus served in this case as the BRET donor. [18,19]. These intramolecular BRET sensors do not simply detect binding of arrestin to a GPCR, but rather its activation state. This is a crucial difference, as arrestin remains in its activated conformation even after its interaction with the GPCR [20].

These described assays have all the disadvantages that both the GPCR and the arrestin have to be genetically modified with fluorescent proteins or other reporter entities (e.g., luciferases), which can affect their function. The issue of tagging can be partially solved by an indirect approach. In this case, a green fluorescent protein (EGFP) was fused either to a polyprenylation domain, localizing it to the plasma membrane, or to an FYVE domain, localizing the EGFP to early endosomes. Expressed together with a luciferase tagged arrestin, activation of the GPCR-arrestin complex can be observed, even if the GPCR remains untagged [21].

Many studies based on luminescence used *Renilla* and firefly luciferases. They have the advantage that the color of the emitted light can be varied by different substrates or the species from which the luciferase originates [22,23]. NanoLuc is a bright, small luciferase, that was originally isolated from *Oplophorus gracilirostris*. This luciferase can be split into two parts: 11S and 114. Both parts are inactive on their own, but when brought into close proximity, their enzymatic activity is restored, leading to luminescence in the presence of a substrate (Figure 1a and Figure 2a). The complementation of the two subunits is reversible, allowing the detection of protein interaction kinetics or transient protein complexes [24,25].

Here we show an enzyme complementation assay based on the split NanoLuc that enables time-resolved monitoring of recruitment of all four human arrestins to both ligand- and light-stimulated GPCRs. This allows the characterization of both receptor selectivity towards arrestins and interaction dynamics. Additionally, an indirect measurement setup allows studying untagged or endogenously expressed GPCRs. BRET- and protein complementation-based systems require the coexpression of at least two components. For this, we used multigene expression systems that allow expression from a single plasmid with different promoters for each expression cassette [26,27,28].

## 2. Results and Discussion

### 2.1. Direct and Indirect Recruitment of Arrestins to GPCRs

Nanoluc enzyme complementation was previously used to measure β-arrestin-2 recruitment to B2AR and vasopressin receptors. This assay is based on β-arrestin-2 that is modified at its N-terminus with the small subunit of NanoLuc (114) and a GPCR with the large subunit (11S) at the C-terminus [29]. We expanded this strategy by using all four human arrestins (Figure 1). We fused the NanoLuc subunits using flexible glycine-serine linkers of 14 and 16 amino acids for arrestins and GPCRs, respectively. We verified the function of the assay using several GPCRs: the B2AR as a prototypical class A receptor that preferentially binds β-arrestin-2, gastrin-releasing peptide receptor (GRPR) and somatostatin receptor type 2 (SSTR2) as examples of peptide receptors, and human rhodopsin (hRho), one of the best-studied light-sensitive receptors. We detected recruitment of all four arrestins to all tested GPCRs upon ligand stimulation, suggesting that our arrestin probes are able to distinguish between the active and inactive state of all tested GPCRs. The B2AR showed a strong selectivity for β-arrestin-2 over all other arrestins, as was expected from its class A nature (Figure 1b) [30]. GRPR and SSTR2 showed binding of both β-arrestins, indicative of a class B behavior. In addition, they recruited S-arrestin (Figure 1c,d). Rhodopsin was stimulated with light flashes and showed a strong interaction with its native binding partner S-arrestin (Figure 1e) as well as both β-arrestins. Arrestin-C consistently generated weak signals, both before and after stimulation of the receptor (Figure 1f). This may be due to low affinity of arrestin-C for the tested GPCRs or low expression of the construct.

It is notable that the baseline signal prior to stimulation of the different GPCRs and arrestins are not equal, which may be indicative of constitutive receptor activity. β-arrestin-2 consistently generated the highest baseline values among all four receptors studied. This is not surprising, as β-arrestin-2 was reported to exist in a pre-activated form in contrast to other arrestins, making it much more promiscuous [31].

The kinetics of arrestin recruitment appears mainly determined by the receptor and not by the arrestin. All four arrestins share similar interaction dynamics for a tested GPCR. B2AR is supposed to only transiently interact with arrestins, and indeed, measured receptor–arrestin interactions weaken over the recorded time course of 40 min. SSTR2 and GRPR showed similar interaction with all arrestins over time, as expected form their class B behavior (Figure 1c,d). Rhodopsin showed a very rapid increase in signal, after which the complex decayed with a half-time of 100 to 180 s, depending on the arrestin tested (Figure 1e). This is in line with its biological function: the rapid acquisition of visual information from the environment at high frequency. Rhodopsin uses 11-cis-retinal as a cofactor that is rapidly released after light-induced isomerization to its all-trans form. The resulting rhodopsin apoprotein seems to quickly dissociate from arrestin. Furthermore, we did not detect any difference in baseline signal when all-trans or no retinal was added to the cells instead of 11-cis retinal, and no response was detected after stimulation with light under these conditions (data not shown).

The direct assay (Figure 1a) has the disadvantage that GPCRs are modified with a tag. This has a strong impact on GPCRs that possess a PDZ binding motif at their C-terminus that is relevant for endosomal trafficking and sorting. Instead of fusing the 11S split-luciferase subunit directly to a transmembrane receptor, we attached this subunit to a CAAX polyprenylation motif localizing it to the plasma membrane (Figure 2a). This setup does not directly detect the interaction of arrestins with GPCRs, but rather the translocation of arrestins from the cytoplasm to the plasma membrane. This indirect assay allows the measurement of label-free receptors without possible steric interference of fused protein tags on the receptor and allows studying endogenously expressed GPCRs from primary cells. A similar approach was taken by Cao et al. [32] to establish a BRET-based assay.

We used this assay to determine binding of all four human arrestins to the unlabelled B2AR, which again showed the strong preference of the class A receptor towards β-arrestin-2 (Figure 1a and Figure 2b). Unlabelled SSTR2 and GRPR were also tested (Figure 2c,d). A higher baseline was observed, which could be caused by unspecific arrestin recruitment to the membrane. This leads to a lower signal-to-baseline ratio after stimulation with agonist compared to the direct assay (Figure 1c,d). Stable HEK293 cell lines expressing both the 11S-CAAX membrane sensor and either S-arrestin, β-arrestin-1 or β-arrestin-2 were developed. All cell lines showed increased chemiluminescence after transient transfection of untagged GRPR, demonstrating their functionality (Figure 2e). These three cell lines can be used as a basis for a ligand screen of any untagged GPCR by expressing the receptor either transiently or stably.

We conclude from our findings that all four human arrestins are able to differentiate between the active and inactive conformation of all tested GPCRs in both the indirect and direct assay. The results also reveal strong specificities of all receptors towards certain arrestins. The measured interaction dynamics were determined by the receptor and not the arrestin, indicating that the receptor determines the lifetime of the complex and therefore the speed of desensitization independent of which arrestin is bound.

### 2.2. Comparison of Different Assay Types

The direct and indirect assays were compared by performing concentration–response experiments with a reference agonist of B2AR (isoproterenol), GRPR (AMBA) and SSTR2 (DOTA-TOC) (Figure 3a). The maximum response (Emax) was set to 100% for all tested GPCRs individually since the absolute light emission cannot be compared among GPCRs. The concentration–response curves comparing the direct and indirect assay showed that the assays are comparable (Figure 3a). We obtained similar potency values in both the direct and indirect assay for B2AR (145 ± 10 nM vs. 161 ± 12 nM) and SSTR2 (61 ± 8 nM vs. 80 ± 15 nM). Only for GRPR (583 ± 65 nM vs. 344 ± 63 nM) did we see a significantly (*p* < 0.05) increased potency in the indirect assay compared to the direct assay (Figure 3a). This might be explained by the differences of these two assays. With the indirect assay, we only detect receptors–arrestin interactions specifically at the plasma membrane and not during the internalization process [21,33]. Therefore, the GRPR probably behaves differently during the internalization process compared to the other tested receptors.

To further validate the direct assay, we compared the recruitment efficacies of different β-arrestin-2 mutants to the B2AR in the direct assay and a classical BRET-based approach and compared the results obtained from both assays. The function of these arrestin mutants was described before [34,35]. No significant difference between the results of the two assays could be observed, providing evidence that the direct assay reports accurate measurements for relative binding strength (Figure 3b). All assay types lead to similar results, suggesting that they really measure the GPCR–arrestin interaction and not artifacts of the detection system.

### 2.3. The C-Tail of GPCRs Modulates Arrestin Selectivity

GPCRs have two binding sites for arrestins. The first site is composed of cytoplasmic loops, mostly the intracellular loop 3 that interacts with the central crest of arrestin, and also a cavity created in the transmembrane bundle by the outward movement of helix V and VI after ligand binding. This is often referred to as the “core interaction” [36]. This interaction occupies a similar binding area as the α-subunit of the G protein, leading to a direct competition between arrestins and G proteins for active GPCRs [37,38,39]. The second binding, called the “tail interaction”, occurs between the cytosolic tail of a GPCR and N-terminal domain of arrestin [40] and has been shown to modulate arrestin activity [41]. Molecular simulations indicated that both sites are able to activate arrestin independently, and binding of arrestin to either core-deficient or C-tail truncated GPCRs could be shown [42].

To investigate the dynamics of the core binding site of arrestin, we created C-tail truncated versions of both B2AR (B2ARΔC) and GRPR (GRPRΔC) [43,44]. β-arrestin-2 showed a stronger recruitment shortly after agonist addition to both the B2ARΔC and GRPRΔC (Figure 4a,b). However, this enhanced recruitment is rapidly lost within 10 min after stimulation. The B2ARΔC declines to recruitment levels equal to those of the full-length receptor after 10 min. For GRPRΔC, the loss of recruitment is much more pronounced with the signal plummeting to less than 50% of the initial response within 10 min after stimulation. This suggests that the unphosphorylated tail of a GPCR acts as an inhibitor in an early phase of recruitment of β-arrestin-2 and that in this phase the binding is based on the core interaction. In a later phase, especially GRPR relies on its C-tail.

β-arrestin-1 binding showed receptor-specific effects: in the case of the B2AR, binding to the C-tail deficient variant was increased compared to full-length receptor during the duration of the assay, while for the GRPR, β-arrestin-1 binding to the truncated version was reduced (Figure 4a,b). This suggests that the C-tail of B2AR prevents β-arrestin-1 recruitment and that this inhibitory function plays a role in the arrestin specificity of class A GPCRs. For GRPR, a class B GPCR, the reduced binding points to an important contribution of the C-tail for β-arrestin-1 recruitment, suggesting a more equal contribution of both binding sites.

### 2.4. Influence of Expression Level on Arrestin Recruitment

Gene dosage experiments were performed to test how the expression levels of the individual binding partners affect arrestin recruitment. We varied expression levels of both GPCR-11S and 114-β-arrestin-2 fusion proteins by transfecting different ratios of plasmids encoding β-arrestin-2 or receptor. The total amount of DNA was kept constant. We observed that the baseline of each GPCR/arrestin complex prior to stimulation depends on expression level ratios (Figure 5a–c, left column). A Job plot was created by plotting the baseline luminescence of all conditions against the fraction of receptor-encoding plasmid used. The highest baseline was observed in the case of a 1-to-1 mixture of plasmids. Two linear fits are shown in each Job plot, each fitted to all values either below or above the GPCR fraction of 0.5. A high binding affinity would cause a nonlinear, flattened curve [45]. The strong linear trends observed indicate a low affinity of β-arrestin-2 binding to the inactive receptors and of the attached luciferase subunits to one another. The low affinity of the 11S to the 114 subunits has been reported previously [29].

For B2AR we observed differences in dynamic behavior after stimulation with agonist, depending on the ratio of plasmids used (Figure 5a, left column). Differences are best visualized when displaying the fold change after stimulation, thereby normalizing to the baseline (Figure 5a, right column). Using an excess of plasmid encoding β-arrestin-2, we see a short, transient response that decays rapidly within 10 min. This time course likely indicates that all receptor on the plasma membrane is internalized by an excess of arrestin within a short time. With a lower amount of β-arrestin-2-encoding plasmid and a high amount of receptor, we see a prolonged response that persists and decays slowly over the recorded time span. The prolonged signal is likely due to the continuous recruitment of arrestin due to a large number of receptors on the plasma membrane. SSTR2 shows a similar behavior (Figure 5b). In contrast, GRPR shows no change in its dynamic behavior regardless of the transfection condition used (Figure 5c). This is possibly due to a prolonged interaction of GRPR with β-arrestin-2, even after internalization.

Transient cotransfections lead to a heterogeneous cell population with different expression ratios in individual cells. To overcome this problem, we expressed both expression cassettes from the same plasmid, which leads to a more constant expression ratio between cells [26]. In addition, expression of each cassette is driven by a separate promoter. We used either the strong cytomegalovirus (CMV) promoter or the approximately 15-fold weaker phosphoglycerate kinase (PGK) promoter [27]. Constructs were created containing the GRPR as well as either β-arrestin-1 or -2. All possible promoter combinations were tested (Figure 5d,e, right column). Constructs expressing both cassettes under control of the CMV promoter showed the highest signal, but the baseline was also clearly increased (Figure 5d,e, left column). Using the PGK promoter in one or both expression cassettes led to a decreased luminescence. However, absolute signal strength is not the only determinant of an assay’s output. A large signal-to-baseline change is desirable especially when different ligands are compared, for example, in a ligand screen for pharmaceutical measurements. Expressing arrestin under the PGK promoter and GRPR under the CMV promoter decreases the fold change after stimulation, as previously shown in the gene dosage experiment (Figure 5c, right column), while expressing both cassettes under the PGK promoter leads to an increased signal fold change after stimulation.

### 2.5. Transduction of Cells with Baculoviruses

Transfection of plasmids into mammalian cells, especially primary cells, is usually an inefficient process. Baculoviruses are attractive vectors for transduction because they transduce mammalian cells with high efficiency and little toxicity. For our purpose, they are well suited because they can carry multiple expression cassettes [26,27].

First, we used a virus that expresses B2AR-11S and 114-β-arrestin-1 or -2 (Figure 6a). A clear increase in signal after stimulation of B2AR with isoproterenol can be seen. As observed before, the arrestin recruitment response for β-arrestin-2 is much stronger than for β-arrestin-1. Next, we transduced the components of the indirect assay consisting of either B2AR, as well as 11S-CAAX and 114-β-arrestin-1, or 2 using a single virus (Figure 6b). The observed response after stimulation indicates successful transduction of the HEK293 cells with up to three different constructs from a single viral vector. Importantly, we showed the recruitment of both β-arrestins to untagged GPCRs using baculovirus transduction. This would allow the measurement of arrestin recruitment in primary cells expressing endogenous GPCRs.

## 3. Conclusions

We successfully developed and verified arrestin recruitment assays for all four human arrestins, showing their ability to bind to a variety of both ligand- and light-stimulated GPCRs. Our results reveal that the expression levels of GPCRs and arrestins play a crucial role for signal strength and kinetics. The assays can be further improved by controlling expression levels, e.g., from a single plasmid with different promotors. For an appropriate modeling of signaling processes in the future, methods for measuring intracellular protein concentrations have to be included. These concentrations can be obtained either by Western blotting or quantitative mass spectroscopy. Alternatively, GPCRs and arrestins could be labeled by an additional fluorophore as a measure for their expression level.

## 4. Materials and Methods

### 4.1. Molecular Biology

Coding regions of proteins were obtained as geneblocks from IDT (Integrated DNA Technologies, Inc., Coralville, IA, USA). Fragments were amplified by PCR using Phusion^®^ High-Fidelity DNA Polymerase (HF or GC, NEB, Cat. No M0530). Sequences of geneblocks and primers are given in Appendix A. Fragments were digested using SapI (or its isoschizomer LguI), separated using TAE agarose gels and then purified using a QIAquick gel extraction kit (Qiagen, Cat. No.: 28104, Düsseldorf, Germany). PCR fragments were cloned into plasmids pSI-AGR10, pSI-AKR1, pSI-DSZ2cx, or pSI-DAZ2, which have been described previously [26,27]. The integrity of all constructs was tested by restriction mapping and sequencing. The GFP-β-Arr-2 coding sequence was kindly provided by Hans Bräuner-Osborne (Copenhagen, Denmark) and was re-cloned into a pEGFP (Clonetech) expression vector. The β2AR-RLuc8 plasmid was a kind gift of Nevin Lambert (Augusta University, Georgia).

### 4.2. Cell Culture

Human Embryonic Kidney (HEK293) cells were cultured in DMEM with high glucose (Bioconcept, Allschwil, Switzerland, Cat. No 1-26F03-I) containing 10% fetal calf serum (FCS). Cells were incubated at 37 °C in a humidified atmosphere containing 5% CO_2_. Transfections were performed in 6-well format using Lipofectamine 3000 transfection reagent (ThermoFisher, Cat. No L3000015, Waltham, MA, USA) according to the manufacturer’s recommendations and using 2 μg of DNA for each well. For gene dosage experiments, one of the encoding plasmids was diluted with carrier DNA so that the concentration of the other plasmid in the transfection mix was kept constant. The molar fraction was calculated by the amount of GPCR-encoding plasmid used divided by the total amount of transfected DNA (excluding carrier). The combinations of plasmids used for transfection can be found in Appendix A.

### 4.3. Split Luciferase Assay

HEK293 cells were cultured in DMEM high glucose (Bioconcept, Cat. No 1-26F03-I, Allschwil, Switzerland) containing 10% FCS. Twenty-four hours after transfection, cells were seeded into a 96-well format with 80,000 cells in each well (Perkin Elmer, #6005181, Waltham, MA, USA). Twenty-four hours after seeding (48 h after transfection) the media was exchanged to DMEM w/o phenol red (Bioconept Cat. No 1-26F23-I, Allschwil, Switzerland), supplemented with 20 mM HEPES pH 7.0, 1% furimazine and 19% LCS dilution buffer (Promega, Nano-Glo^®^ Live Cell Assay System, Cat. No N2012, Madison, WI, USA). A white cover was attached to the transparent bottom of the 96-well plate. Luminescence was measured using a PHERAstar FSX (BMG Labtech, Offenberg, Germany). Cells were stimulated with 1 µM of agonist unless noted otherwise. To measure opsins, 1 µM of *11cis*-retinal was added to the cell media after seeding the cells into a 96-well format. During and after retinal addition, cells were always protected from light of wavelengths below 600 nm. Stimulation of opsins was performed with the built-in 20 mW flashlamp of the PHERAstar, delivering 500 flashes lasting 2 µs each over a time of approximately 2 s to each 96-well, with a wavelength centered at 485 nm (+/−5 nm). Luminescence time courses are representative experiments shown as mean ± SD of at least three replicates.

### 4.4. BRET Assay

The arrestin recruitment assay is based on bioluminescence resonance energy transfer (BRET). To prepare BRET, donor and acceptor constructs β-arrestin-2 and B2AR were modified as followed: B2AR was C-terminally tagged with RLuc8 (BRET donor) and the β-arrestin-2 WT and mutants were N-terminally tagged with GFP (BRET acceptor). The luciferase RLuc8 catalyzes oxidation of its substrate (Coelenterazine 400A), which results in the emission of photons (λmax = 410 nm), which will excite the GFP if it is in close proximity. HEK293 cells were transiently co-transfected with BRET donor and BRET acceptor constructs with Lipofectamine2000 according to manufacturer’s protocol. Briefly, HEK293 cells were seeded at a density of 500k cells/well in a cell culture 6-well plate; 24 h later, cells were transfected using 2 µg total DNA, 6 µL Lipofectamine2000, 150 µL OptiMEM/6-well. The ratio of BRET acceptor construct to BRET donor construct was 1:10. The cells were transferred into a poly-L-Lysine-coated, white, sterile 96-well microplate at a density of 50 k/well 24 h after transfection and cultured overnight. Cells were incubated in 90 µL HBSS (supplemented with 20 mM HEPES, pH 7.4 and 50 µM Coelenterazine 400A), and baselines were measured on a PHERAstar FSX from BMG Labtech (Ortenberg, Germany). Cells were stimulated with isoproterenol from 0.01 nM to 10 µM. The isoproterenol-induced recruitment of GFP-β-arrestin-2 WT and mutants to the β2AR-RLuc8 brings GFP into close proximity of the BRET donor, which results in its excitation and emission of light at 515 nm. BRET ratios (515 nm/410 nm) were detected with the optic module BRET2 plus (515–530 nm, 410–480 nm) and plotted as a function of time. Emax values were calculated from concentration–response curves with area under the curve (AUC) vs. ligand concentrations plotted in GraphPad Prism 8. The concentration–response curves with area under the curve (AUC) vs. ligand concentrations were fitted using the nonlinear regression “log (inhibitor) vs. response (three parameters)” in GraphPad Prism 8 to calculate the Emax values. Emax values for GFP-β-arrestin-2 mutants were normalized to the Emax value of GFP-β-arrestin-2 WT recruitment.

### 4.5. Data Analysis

Data shown are either the mean ± SD of three replicates measured in a single experiment or mean ± SEM of at least three different experiments. Fold change was calculated by dividing all measured values from a single well by the last value recorded before stimulation. The results of several wells are then displayed as mean ± SD. Area under a curve was calculated by subtracting the time courses of unstimulated wells from that of all stimulated wells. Net area was calculated using GraphPad Prism.

### 4.6. Baculovirus Generation

Baculovirus used to infect mammalian cells was produced as described previously (Mansouri et al., 2016) [27]. In brief, electrocompetent *E. coli* DH10EMBacY cells were transformed with the plasmids described in Appendix A. Positive colonies were identified in a blue-white screening. Bacmids were purified from liquid culture using a ZR BAC DNA kit (Zymo Research, Cat No D4048). Sf21 insect cells seeded into 6-well plates and grown at 27 °C were transfected with the bacmid using the CellFECTIN II transfection reagent (Thermo, Cat No 10362100). The virus was propagated in Sf21 cells over several incubation steps at 27 °C and constant agitation lasting eight days in total and was harvested using centrifugation. The resulting supernatant was concentrated using ultrafiltration units (Vivaspin 20, 100,000 MWCO PES, 48pc Cat. No VS2042) and then used to transduce HEK293 cells.

## Figures and Tables

**Figure 1 ijms-21-04949-f001:**
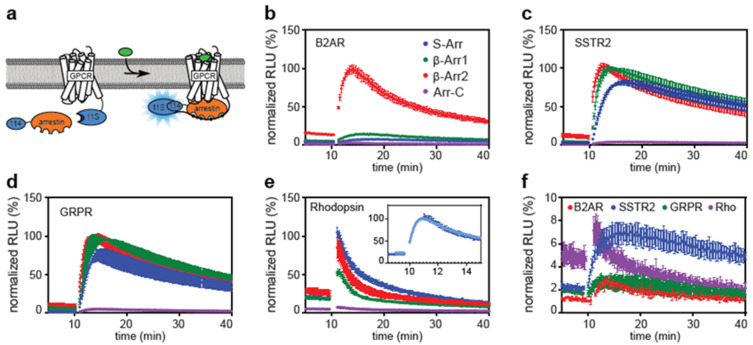
Direct arrestin recruitment assay showing the recruitment of all four human arrestins to various G protein-coupled receptors (GPCRs). (**a**) Principle of the assay. The large subunit of NanoLuc (11S) is expressed as a fusion protein with the GPCR. The small subunit (114) is linked to the N-terminus of arrestin. Upon GPCR stimulation, arrestin is recruited to the GPCR, and NanoLuc is reconstituted and becomes active. After 10 min of baseline recording, GPCRs were stimulated with 1 µM of their respective agonist (isoproteronol, DOTA-TOC, AMBA), or in the case of rhodopsin with light of 485 nm. (**b**) B2AR mainly recruits b-arrestin-2 (β-Arr2). (**c**–**e**) Somatostatin receptor type 2 (SSTR2), GRPR and rhodopsin recruit S-arrestin (S-Arr), β-arrestin-1 (β-Arr1) and β-arrestin-2 at similar levels. (**e**) Rhodopsin recruits arrestin faster than other GPCRs recruit their respective ligands. The inset shows data from the standard data collection method (dark blue) and one with a higher temporal resolution (light blue). (**f**) Arrestin-C (Arr-C) recruitment is lower, but similar kinetics as for other arrestins was observed. Shown are time courses of the mean ± SD of a representative experiment. Data were normalized to the maximum response generated by a given GPCR.

**Figure 2 ijms-21-04949-f002:**
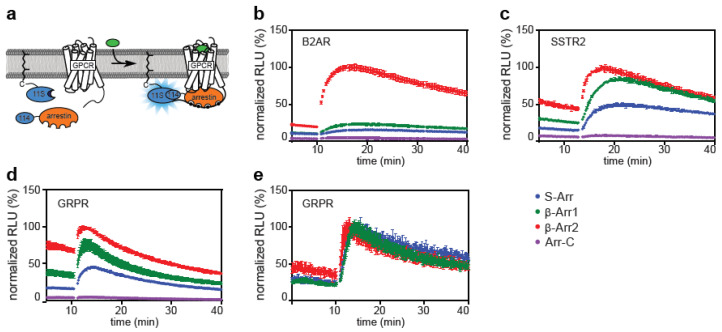
Indirect arrestin recruitment assay showing the recruitment of all four human arrestin isoforms to various GPCRs. (**a**) Principle of the assay. The large subunit of NanoLuc is expressed with a prenylation signal (11S-CAAX) localizing it to the plasma membrane. The small subunit (114) is linked to the N-terminus of arrestin. These components are either transiently transfected (**b**–**d**) or stably expressed (**e**) in HEK293 cells. Upon stimulation with ligand, arrestin is recruited to GPCRs at the plasma membrane and chemiluminescence is enhanced. (**b**) B2AR mainly recruits β-arrestin-2 (β-Arr2). (**c**,**d**) SSTR2 and GRPR recruit S-arrestin (S-Arr), β-arrestin-1 (β-Arr1) and b-arrestin-2. Recruitment of arrestin-C (Arr-C) is low with all tested GPCRs. (**e**) Stable HEK293 cell line expressing 11S-CAAX and S-arrestin, β-arrestin-1 or β-arrestin-2. Untagged GRPR was transiently cotransfected. Data collected from each receptor were normalized to the maximum signal generated from the recruitment of β-arrestin-2 to that receptor. For the stable cell lines data from each time course was normalized to the maximum response observed. Shown are time courses of the mean ± SD of a representative experiment.

**Figure 3 ijms-21-04949-f003:**
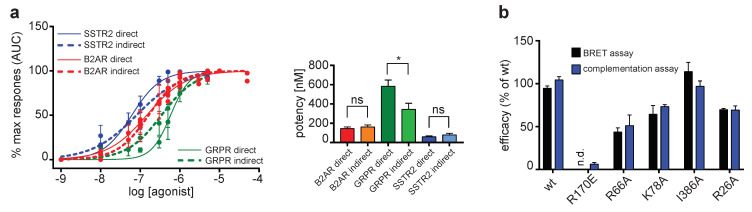
Comparison of different arrestin recruitment assays. (**a**) β-arrestin-2 recruitment was measured for B2AR, GRPR and SSTR2 stimulated with different concentrations of their respective agonist in both the direct and indirect assay approach. Dose response was determined by calculating the area under the curve (AUC) from the time point of agonist addition to the end of the experiment. All responses were normalized to the maximum response generated by each receptor. Data from three independent experiments ±SEM are shown. Potency was calculated from dose–response curves for each assay type using a four-parameter nonlinear regression and compared using an unpaired t-test. *: *p* < 0.05, ns: not significant. (**b**) Comparison of NanoLuc complementation and a bioluminescence resonance energy transfer (BRET)-based arrestin recruitment assays. For the NanoLuc complementation assay, efficacies were calculated from concentration–response curves using isoproteronol in at least three independent experiments and normalized to β-arrestin-2. For the BRET arrestin recruitment assay, HEK293 cells transiently co-expressing β2AR-RLuc8 and GFP-βArr2 WT and mutant constructs were treated with the luciferase substrate coelenterazine 400A and stimulated with increasing concentration of isoproterenol. The area under curve (AUC; *t* = 0–20 min) was plotted as a function of isoproterenol concentrations, and normalized Emax values are plotted (black bar). For βArr2-R170E, it was not possible to determine an Emax value (n.d.). Data represent mean ± SEM of two to six individual experiments carried out in duplicate. No significant differences in efficacies between the two assay types were observed.

**Figure 4 ijms-21-04949-f004:**
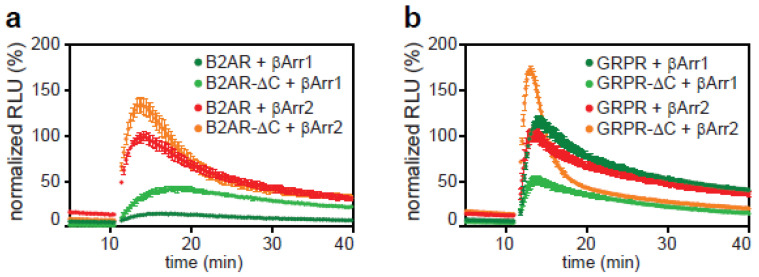
Core interaction with a GPCR is sufficient for arrestin recruitment. (**a**) C-tail truncated variants of the β2AR and GRPR were compared to their wild-type forms using the direct split luciferase assay. (**a**) Recruitment of either β-arrestin-1 or -2 to full-length B2AR-11S or B2AR-11S with a deletion of the cytoplasmic tail (B2AR-∆C). (**b**) Recruitment to full-length GRPR-11S or GRPR-11S without cytoplasmic tail (GRPR-∆C). Data collected for each receptor were normalized to the maximum signal generated from the recruitment of β-arrestin-2 to the wild-type form of that receptor. Shown are time courses of the mean ± SD of a representative experiment.

**Figure 5 ijms-21-04949-f005:**
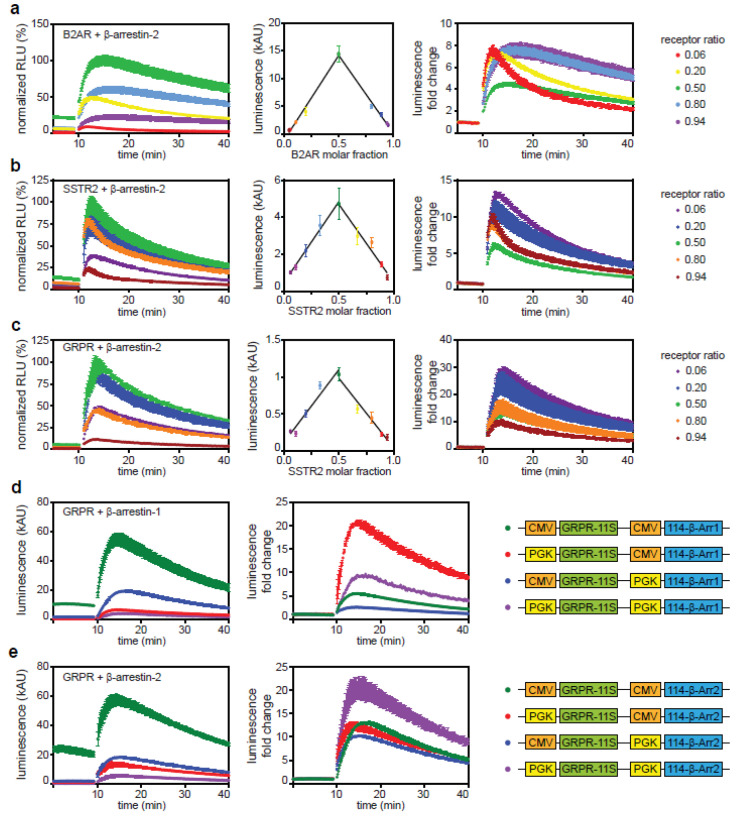
Influence of expression levels. (**a**–**c**) Gene dose experiments. Different ratios (indicated on the right side of the panels) of expression plasmids for Β2AR-11S, GRPR-11S or SSTR2-11S were transiently co-transfected with an expression construct for 114-β-arrestin-2 in HEK293 cells. Cells were stimulated with 1 µM of the respective agonist after 10 min of baseline measurement. Data were normalized to the maximum response generated (left). Job plots were generated by plotting the recorded baseline luminescence against the molar fraction of receptor (middle). Fold change of activation was calculated by dividing each signal by the corresponding baseline (right). Shown are time courses of the mean ± SD of a representative experiment. (**d**,**e**) MultiPrime plasmids encoding both the GRPR-11S and 114-β-arrestin-1 or -2 were transiently transfected into HEK293 cells. Two different promoters were used to control the expression levels of the biosensors: cytomegalovirus (CMV) and phosphoglycerate kinase (PGK) promoter (right). After 10 min of baseline measurement, transfected cells were stimulated with 1 µM of AMBA. Data are either normalized to the maximum response generated (left, **a**–**c**), raw chemiluminescence (left, **d**,**e**) or fold change. Shown are time courses of the mean ± SD of a representative experiment.

**Figure 6 ijms-21-04949-f006:**
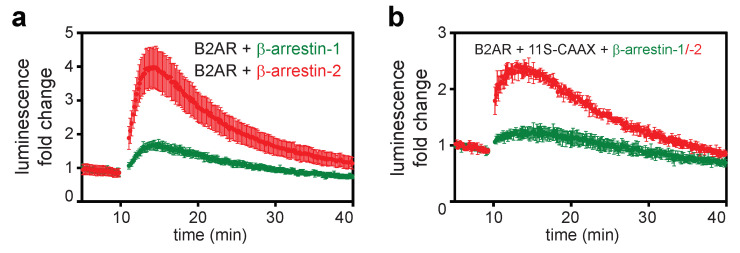
Transduction of mammalian cells with arrestin recruitment reporter systems. (**a**) Transduction of HEK293 cells with a baculovirus expressing B2AR-11S and 114-β-arrestin-1 or 114-β-arrestin-2. (**b**) Transduction of HEK293 cells with a baculovirus expressing untagged B2AR, 11S-CAAX and 114-β-arrestin-1 or 114-β-arrestin-2. In both assays, preferential recruitment of β-arrestin-2 was observed as for the assays with transient transfection. Shown are time courses of the mean ± SD of a representative experiment. Data were normalized to the maximum response generated with each virus.

## Data Availability

Data and plasmids are available upon request.

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
