# Peer review of "New Insights into Arrestin Recruitment to GPCRs"

_ijms, 2020, doi:10.3390/ijms21144949_

Round 1
Reviewer 1 Report
The manuscript by Spìllmann et al. describes a novel methodological approach to investigate arrestings recruitment to GPCRs. The authors propose and indirect method that avoids tagging of the GPCRs at their C-terminal tails which are known to be important signalling domains. This is an important issue because the C-terminus of GPCRs can be involved in trafficking and even in other important steps like G-protein binding/recognition process. The authors show the usefulness of this indirect approach by analyzing results with four distinct arrestins and four relevant GPCRs. Their results are compared to the direct and other relevant methodologies used in the literature. On the basis of their results, they can also propose that the C-terminus of GPCRs is a potential modulator of arrestin binding selectivity. Futhermore, the authors analyze the effect of the expression ratios of arrestin/GPCR to evaluate the effect on the observed luminiscent interacting signal.
The manuscript is well written and the experiments are conducted to a good scientific standard with the corresponding controls and statistical analysis. Overall, the conclusions are backed up by the results presented. The proposed methodology can be useful to the GPCR community to dissect in more detail GPCR-arrestin interactions avoiding the potential undesired effects of direct methods that tag the C-terminus of the GPCR. There are, however, several aspects that the authors should discuss and improve for a better clarity of the presented results.
1) The initial study includes the visual GPCR rhodopsin and the authors state in lines 168-173 that the direct and indirect approches have been validated for all the arrestins and GPCRs studied. However, the visual GPCR rhodopsin is only shown on Figure 1 but does not appear in the indirect assay of Figure 2 and others. Is there any reason for not including rhodopsin in these cases? Please comment.
2) The legend to Figure 2 could be better explained. There are apparently only two panels (a and b) but the legend refers to c. Also the description of the mutants is in the text but not in the Figure legend. Please check this.
3) Figure 5 is a complex figure. The figure legend and labels could be improved for better clarity. Particularly, the text lines 305-311 (effect of the different CMV and PGK promotors) is not clear to me that it matches the behaviour observed in the figure panels. Please check consistency and try to improve the layout/explanation of the different figure panels.
4) The effect of the expression levels on the described methodology is difficult to rationalize. The authors could discuss this in further detail.
5) Related with 4): although the difficulty is acknowledged, is there any possibility to get quantitative results from the methodology reported? What should be the requirements for this to be possible? This can be discussed.
Author Response
Reviewer 1 Point 1: The initial study includes the visual GPCR rhodopsin and the authors state in lines 168-173 that the direct and indirect approches have been validated for all the arrestins and GPCRs studied. However, the visual GPCR rhodopsin is only shown on Figure 1 but does not appear in the indirect assay of Figure 2 and others. Is there any reason for not including rhodopsin in these cases? Please comment.
Reply: Unfortunately, we never used rhodopsin with the indirect assay. We started to work with light-activated GPCRs in the context of a different project and we used rhodopsin as a starting point as control. We decided to include rhodopsin data of the direct assay in this manuscript to show the potential of the assay even it appears now inconsistent.
Reviewer 1 Point 2:The legend to Figure 2 could be better explained. There are apparently only two panels (a and b) but the legend refers to c. Also the description of the mutants is in the text but not in the Figure legend. Please check this.
Reply: This comment refers to Figure 3. We would like to apologize for the mistakes in this figure. We also updated the data of Figure 3c and adjusted the figure legend. The message of the figure remains the same. We added refences in the text that link to the functional background of these mutants (Line 218).
Reviewer 1 Point 3/4: Figure 5 is a complex figure. The figure legend and labels could be improved for better clarity. Particularly, the text lines 305-311 (effect of the different CMV and PGK promotors) is not clear to me that it matches the behaviour observed in the figure panels. Please check consistency and try to improve the layout/explanation of the different figure panels.
The effect of the expression levels on the described methodology is difficult to rationalize. The authors could discuss this in further Detail.
Reply: We made several changes in the figure legend and in the text and we hope that it is now clearer. (Line 312-324; Line 326-333)
Reviewer 1 Point 5: Related with 4): although the difficulty is acknowledged, is there any possibility to get quantitative results from the methodology reported? What should be the requirements for this to be possible? This can be discussed.
Reply: We added two sentences at the end in the “Conclusions” part. This shows the future direction of this assay type.
Reviewer 2 Report
This is a very nice technical paper that includes some new research findings. I think that this manuscript highlights some new techniques that will be useful to the GPCR community.
I have only two minor comments:
- Line 214: rather than stating "than the other tested receptors," appropriate wording should be "compared to the other tested receptors."
- Line 219: remove the word "Conclusively"
Author Response
Reviewer 2:
Line 214: rather than stating "than the other tested receptors," appropriate wording should be "compared to the other tested receptors."
Line 219: remove the word "Conclusively"
Reply: Both changes were made (Lines 214, 220)
Reviewer 3 Report
I think the MS is well written, novel and of high scientific quality and deserves publication in IJMS.
Author Response
No changes were necessary